# Technical note: A sensitivity analysis from 1 to 40 GHz for observing the Arctic Ocean with the Copernicus Imaging Microwave Radiometer

Lise Kilic[1], Catherine Prigent[1,2], Carlos Jimenez[2,1], and Craig Donlon[3]

[1]Sorbonne Université, Observatoire de Paris, Université PSL, CNRS, LERMA, Paris, France
[2]Estellus, Paris, France
[3]European Space Agency, Noordwijk, the Netherlands

**Correspondence:** Lise Kilic (lise.kilic@obspm.fr)

**Abstract.** The Copernicus Imaging Microwave Radiometer (CIMR) is one of the high priority missions for the expansion of the Copernicus program within the European Space Agency (ESA). It is designed to respond to the European Union Arctic policy. Its channels, incidence angle, precisions, and spatial resolutions have been selected to observe the Arctic Ocean with the recommendations expressed by the user communities. In this note, we present the sensitivity analysis that has led to the choice of the CIMR channels. The famous figure from Wilheit (1979), describing the frequency sensitivity of passive microwave satellite observations to ocean parameters, has been extensively used for channel selection of microwave radiometer frequencies on board oceanic satellite missions. Here, we propose to update this sensitivity analysis, using state-of-the-art radiative transfer simulations for different geophysical conditions (Arctic, mid-latitude, Tropics). We used the Radiative Transfer Model (RTM) from Meissner and Wentz (2012) for the ocean surface, the Round Robin Data Package of the ESA Climate Change Initiative (Pedersen et al., 2019) for the sea ice, and the RTM from Rosenkranz (2017) for the atmosphere. The sensitivities of the brightness temperatures (TBs) observed by CIMR as a function of Sea Surface Temperature (SST), Sea Surface Salinity (SSS), Sea Ice Concentration (SIC), Ocean Wind Speed (OWS), Total Column Water Vapor (TCWV), and Total Column Liquid Water (TCLW) are presented as a function of frequency between 1 to 40 GHz. The analysis underlines the difficulty to reach the user requirements with single channel retrieval, especially under cold ocean conditions. With simultaneous measurements between 1.4 and 36 GHz onboard CIMR, applying multi-channel algorithms will be facilitated, to provide the user community with the required ocean and ice information under arctic environments.

## 1 Introduction

The Copernicus Imaging Microwave Radiometer (CIMR) is currently being implemented by the European Space Agency (ESA) as a High Priority Copernicus Mission (HPCM). It partly follows previous studies conducted at ESA for the Multifrequency Imaging Microwave Radiometer (MIMR) (Bernard et al., 1990). CIMR will deploy a wide-swath ($>1900$ km) conically scanning multi-frequency microwave radiometer with a 55° incidence angle with the Earth surface. CIMR measurements will be made using a forward scan arc followed ~260 seconds later by a second measurement of the same location using a back-

**Table 1.** CIMR characteristics as expressed in Donlon and CIMR Mission Advisory Group (2020).

| Frequency (GHz) | Spatial resolution (km) | Incidence angle (°) | NeΔT (K) |
|---|---|---|---|
| 1.414 | <60 | 55 | 0.3 |
| 6.925 | ≤15 | 55 | 0.2 |
| 10.65 | ≤15 | 55 | 0.3 |
| 18.7 | ≤5 | 55 | 0.3 |
| 36.5 | ≤5 | 55 | 0.7 |

ward scan arc. Polarised (H and V) channels centred at 1.414, 6.925, 10.65, 18.7 and 36.5 GHz are included in the mission design under study. The frequency selection for a satellite mission has to account for the International Telecommunication Union (ITU) frequency regulation, and to ensure continuity with past and current missions. Therefore, the flexibility to choose the channel frequencies and their bandwidths is limited. The real-aperture resolution of the 6.9/10.65 GHz channels is <15 km, and 5 and 4 km for the 18.7/36.5 GHz channels respectively. The 1.4 GHz channel will have a real-aperture resolution of <60 km (fundamentally limited by the size of the ∼8 m deployable mesh reflector) (see Table 1). However, most channels will be oversampled by ∼20% allowing gridded products to be generated at better spatial resolution. Channel NeΔTs are within 0.2-0.8 K with an absolute radiometric accuracy goal of ≤0.5 K. CIMR will fly in a dawn-dusk orbit providing, with one satellite, ∼95% global coverage every day, better than daily coverage poleward of 55° N and S, and will fully cover the poles (no gap). CIMR will operate in synergy with the EUMETSAT MetOp-SG(B) mission so that in the polar regions (>65°N and 65°S) collocated and contemporaneous measurements between CIMR and MetOp MicroWave Imager (MWI)/Ice Cloud Imager (ICI) and SCAtterometer (SCA) measurements will be available within +/− 10 minutes.

CIMR is primarily designed to observe the Arctic environment. Among other parameters, it will provide estimates of the Sea Ice Concentration (SIC), the Sea Surface Temperature (SST), thin Sea Ice thickness (tSIT), Sea Ice Drift (SID), Sea Ice Type, Sea Surface Salinity (SSS), and a range of terrestrial products under clear and cloudy conditions (e.g. soil moisture, permafrost, vegetation dynamics, snow water equivalent). An initial CIMR retrieval capability has been evaluated in Kilic et al. (2018).

One of the key issues to obtain the best precisions on the retrieved parameters is the sensitivity to the parameters to be retrieved. In 1979, Wilheit illustrated the relative sensitivity of the passive microwaves to the ocean parameters for the Scanning Multi-channel Microwave Radiometer (SMMR) (see Figure 1). This figure is certainly the most famous illustration in the ocean passive microwave remote sensing community: it has been reused at many occasions to justify the frequency selection for a large range of missions (e.g., Imaoka et al. (2010); Gabarro et al. (2017)). However, this figure has not been recalculated for a quantitative exploitation of the results. Here, we update the original figure using state of the art Radiative Transfer Models (RTMs) adapted for the range of frequencies used with CIMR. We perform a sensitivity analysis of the passive microwaves to the ocean and ice parameters including the sensitivity to atmosphere, to produce a new key figure useful for the next generation of passive microwave radiometers. This is used to confirm the selection of centre frequencies used by the CIMR mission for different geophysical conditions.

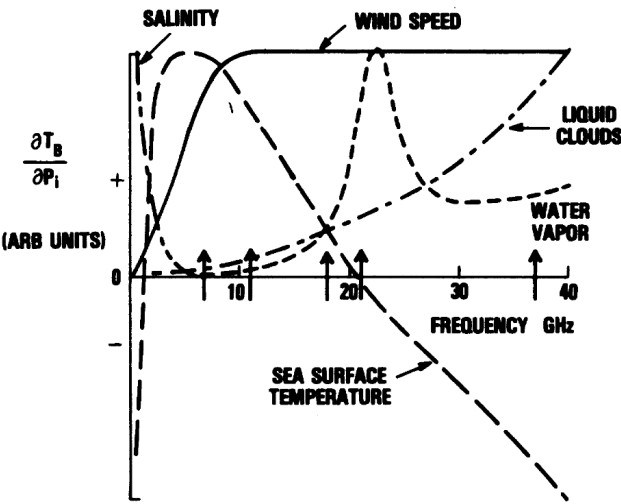

**Figure 1.** Figure 3 in Wilheit (1979). Schematic superposition of the spectra of various geophysical parameters, $P_i$. The arrows indicate SMMR frequencies. The signs have been chosen to be positive in the frequency range of primary importance to the given parameter.

In Section 2, the methodology and the RTMs used to perform the simulations will be described. The results will be presented in Section 3. The sensitivities for a general case corresponding to mid-latitude conditions (because of the median values of the geophysical parameters) will be presented, then we will show the impact of changing the geophysical conditions on the sensitivities of the parameters. Then we will especially focus on the case of the challenging arctic conditions by presenting the sensitivities of the different parameters relative to the CIMR user requirements. Finally, Section 4 will conclude this study.

## 2 Materials and Method

### 2.1 Description of the Radiative Transfer Model

To simulate the sensitivity of the passive microwave satellite observations to the geophysical parameters as a function of frequency over the ocean, a RTM is required. It has to include the simulation of the ocean and ice emissivity, as well as the contribution from the atmosphere, clear and cloudy.

The ocean emissivity varies primarily with the SST, the Ocean Wind Speed (OWS), and the SSS. The emissivity of a flat ocean surface can be simulated from the Fresnel equations, with the sea water permittivity calculated as a function of SST and SSS. When the OWS strengthens, waves appear, the surface gets rougher, and foam can be generated. To calculate the ice-free ocean emissivity, the Remote Sensing Systems RTM (Meissner and Wentz, 2012) is adopted. A comparison of ocean RTMs by Kilic et al. (2019) showed that this model is appropriate for frequencies between 1 and 40 GHz. This model is essentially fitted to satellite observations, with the Special Sensor Microwave/Imager (SSM/I) and WindSat observations between 6-89 GHz (Meissner and Wentz, 2004, 2012), and with Aquarius observations at 1.4 GHz (Meissner et al., 2014, 2018).

Sea ice is a very complex medium composed of different layers of ice, possibly covered by snow. Physically-based emissivity models require a large range of ancillary information that are hardly accessible and they encounter strong difficulties to simulate observations consistently over a large spectral range. The ESA Sea Ice Climate Change Initiative (CCI) Round Robin Data Package (RRDP, Pedersen et al. (2019), https://figshare.com/articles/Reference_dataset_for_sea_ice_concentration/6626549) is a large dataset of co-located brightness temperatures from the Soil Moisture and Ocean Salinity (SMOS) satellite and the Advanced Microwave Scanning Radiometer 2 (AMSR2) over sea ice with relevant meteorological data. Here, the RRDP is used to provide a realistic sea ice emissivity value. The sea ice emissivity varies upon many different parameters (e.g., ice type, ice thickness, snow depth), that can introduce uncertainties. The values and the standard deviations of the ice brightness temperatures at CIMR frequencies estimated from the RRDP are presented in Kilic et al. (2018). The stored brightness and sea ice surface temperatures for the Arctic conditions are extracted and used to derive a representative ice emissivity estimate. The emissivities are first computed at the observation frequencies, which are close to the CIMR observing channels, followed by a smooth interpolation to provide emissivity values at the frequencies between the currently observed channels.

The sensitivities to atmospheric parameters, including Total Column Water Vapor (TCWV) and Total Column Liquid Water (TCLW), are also evaluated. In previous similar studies (Wilheit, 1979; Imaoka et al., 2010), the sensitivities of the signal to the TCWV and the TCLW were estimated, but the atmosphere was not accounted for in the analysis of the sensitivity to the surface parameters. Here, the clear-sky atmospheric contribution is systematically included, leading to more realistic results at the top of the atmosphere, especially above 15 GHz. The widely used RTM of Rosenkranz (2017) for the atmospheric absorption is applied, with the latest improvements in atmospheric gas absorption (Rosenkranz, 1998; Mätzler, 2006; Makarov et al., 2020) as well as a formulation for the cloud liquid water non-scattering contribution (Note that below 40 GHz, hydrometeorological scattering is usually negligible). It includes all the physics required for an accurate evaluation of the atmospheric absorption by water vapor and oxygen in the atmosphere. It is valid from 1 GHz up to 1000 GHz.

## 2.2 Sensitivity computation

The brightness temperatures at the Top-Of-Atmosphere $TB_{TOA}$, at frequency $f$, polarization $p$, and incidence angle $\theta$, is computed as follows:

$$TB_{TOA}(f,p,\theta) = T_s \cdot e(f,p,\theta) \cdot \tau(f,p,\theta) + TB_{down}(f,p,\theta) \cdot \tau(f,p,\theta) \cdot (1 - e(f,p,\theta)) + TB_{up}(f,p,\theta) \tag{1}$$

with $e$ the surface emissivity, $\tau$ the atmospheric transmission, $TB_{down}$ (resp. $TB_{up}$) the atmospheric downwelling (resp. upwelling) brightness temperature. $T_s$ is the surface skin temperature, here a SST or an ice surface temperature ($T_{ice}$). Here a specular reflection is assumed for the ice. As demonstrated in Matzler (2005), the specular approximation is valid for conically scanning instruments such as CIMR with incidence angles close to $55°$. For the ocean, the scattering due to the surface roughness is taken into account by adding a scattering term to $TB_{down}$. This term is given by the Remote Sensing Systems model and depends on $f$, $\theta$, OWS, and the atmospheric opacity.

**Table 2.** Surface and atmospheric conditions for the three considered environments

| Environment | TCWV (kg/m$^2$) | SST ($^\circ$C) | SSS (psu) | OWS (m/s) | $T_{ice}$ ($^\circ$C) |
|---|---|---|---|---|---|
| Arctic | 5 | 0 | 34 | 6 | 0 |
| Mid-latitudes | 20 | 10 | 34 | 6 | - |
| Tropical | 40 | 24 | 34 | 6 | - |

For simplicity, we assume that the extra-terrestrial contributions to the signal (cosmic background, Galaxy, Sun, Moon) as well as the Faraday rotation, have already been removed from the satellite measurements. However, we note that these contributions are especially critical at 1.4 GHz.

Different surface-atmospheric conditions will be considered: mid-latitudes, arctic, and tropical. The TCWV and the SST vary globally between 5 to 70 kg/m$^2$, and 273 to 305 K, respectively, with mean values that strongly depend upon the latitude. The OWS and the SSS vary globally between 0 to 20 m/s, and 32 to 38 psu with mode values around 7 m/s and 34 psu, respectively. For each latitude range / environment, the surface and atmospheric parameters can undergo significant variabilities, here some mean values are chosen for these parameters for illustration purposes. Table 2 summarizes the value of the surface and atmospheric parameters used for each of these environments.

The sensitivity represents the variation of the $TB_{TOA}$ for a given variation of a given parameter. The sensitivities to SST, SSS, OWS, SIC, TCWV, and TCLW are computed using finite differences:

$$K_x = \frac{\Delta TB_{TOA}}{\Delta x} = \frac{TB_{TOA}(x_2) - TB_{TOA}(x_1)}{x_2 - x_1} \tag{2}$$

where $x$ represents the geophysical parameter (SST, SSS, OWS, SIC, TCWV, or TCLW) and $K_x$ the sensitivity to this parameter. Note that we can use the finite difference here to derive the sensitivity, as in the window channels between 1 to 40 GHz at least, the variation of the TB as a function of the parameters is quasi-linear.

In the following, to help interpreting the results, the sensitivity of each parameter from 1 to 40 GHz is normalized by its maximum value (except for Figure 4):

$$K_{x,norm} = \frac{K_x}{max(K_x)} \tag{3}$$

In addition, for each parameter, the most sensitive polarization is selected. It is systematically the vertical polarization for SST and SSS, and the horizontal polarization for the other variables (SIC, OWS, TCWV, and TCLW).

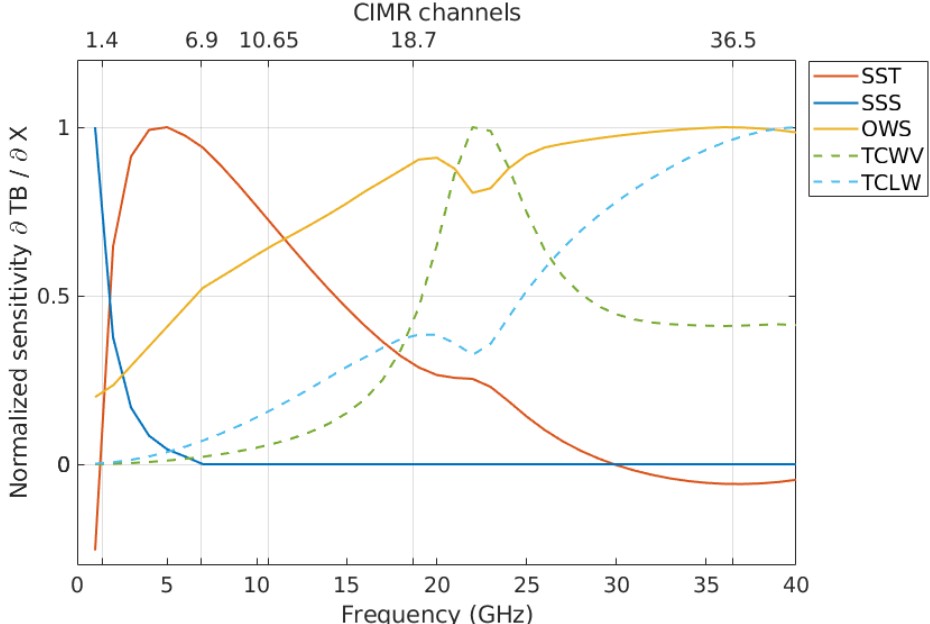

**Figure 2.** Normalized sensitivities of the satellite measurements to surface parameters (solid lines) and atmospheric parameters (dashed lines) as a function of frequency at 55° incidence angle. For the ocean and atmospheric parameters, the mid-latitude conditions are used. The selected central frequencies for CIMR are indicated (vertical bars). Variations of the sensitivities due to the variations of the surface and atmospheric parameters are shown in Figure 3.

## 3 Results

### 3.1 Sensitivity for a general case

Figure 2 shows the normalized sensitivities of the ocean (SST, SSS, OWS), and atmospheric parameters (TCWV, TCLW),
as a function of frequency between 1 and 40 GHz, at 55° incidence angle. The sensitivities are shown for the mid-latitude environment. The CIMR channels are indicated on the top. For each parameter, the sensitivity curve has been normalized as indicated in Eq. 3. Note that the un-normalized sensitivities for the Arctic case will be presented in Figure 4.

    The maximum sensitivity to the SST is obtained at C-band (4-8 GHz) with a value of 0.6 K/K. A large negative sensitivity has been observed on previous similar figures (Wilheit, 1979; Imaoka et al., 2010) at frequencies above $\sim$ 20 GHz: that was
due to the lack of atmospheric contribution in the analysis. For SSS, the sensitivity decreases with frequency above 1 GHz. In the studied range of frequency, the maximum sensitivity for SSS is of 0.7 K/psu at 1 GHz. The sensitivity to OWS is larger at higher frequencies (>18 GHz) with a maximum sensitivity of 0.9 K/(m/s). Note that with this updated version of the Figure of Wilheit (1979) taking into account the sensitivity to the atmosphere, we can see that the sensitivities to the other parameters, such as SST or OWS, are decreased at higher frequencies (>18 GHz), and especially near the water vapor absorption line at
22 GHz.

The CIMR frequencies have been chosen to maximize the measurement sensitivity to the surface parameters, to ensure the continuity of the satellite measurements with current missions (SMOS, SMAP, AMSR2), and to avoid Radio Frequency Interferences (RFI) as much as possible.

The choice of an incidence angle of 55° for CIMR has been constrained by the swath width (to fully cover the poles), and the spatial resolution. By increasing the incidence angle, we increase the swath width, but we degrade the spatial resolution of the measurements for a given satellite altitude (noting that the satellite altitude is fixed to be the same as that of MetOp-SG(B)). This choice of incidence angle has also been tested in terms of sensitivity. The same sensitivity calculations have been performed with smaller incidence angles (not shown): the sensitivities to the ocean surface parameters systematically increase when increasing the incidence angle.

## 3.2 Changes of sensitivity due to the environment

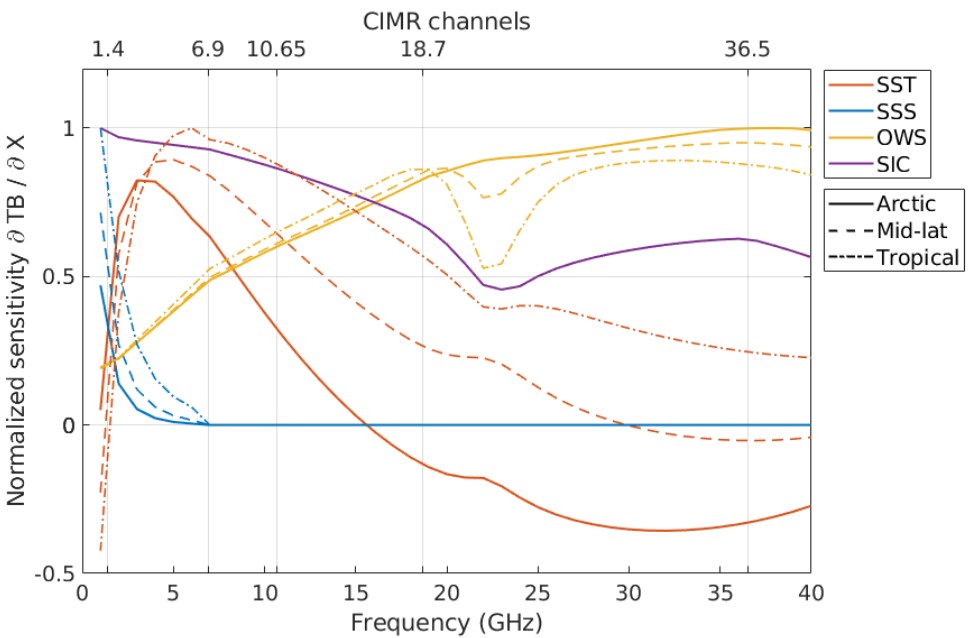

**Figure 3.** Normalized sensitivities of the satellite measurements to the surface parameters as a function of frequency for the different environments at 55° incidence angle, under clear sky conditions. Solid lines: Arctic. Dashed lines: Mid-latitudes. Dotted lines: Tropical.

While CIMR will provide measurements over the global domain on a daily basis, it is primarily designed to observe the arctic environment. This is an extreme environment, with drier atmosphere and colder surface conditions that impact the sensitivity of the satellite observations to the ocean and ice geophysical parameters. This places high demands on the sensitivity of the CIMR radiometer design requirements (Donlon and CIMR Mission Advisory Group, 2020).

Figure 3 shows the sensitivities of the satellite measurements to SST, SSS, OWS, as a function of frequency for arctic, mid-latitude, and tropical environments (the SIC sensitivity is only shown for the arctic conditions). For each ice-free ocean variable, the curves are normalized by the maximum value for the three considered environments.

  The maximum sensitivity to the SIC is provided by the frequencies below 10 GHz with a maximum sensitivity of 1.7 K/1%SIC. Note nevertheless that the SIC retrieval also requires high spatial resolution, with the smaller frequencies suffering from a

150 coarser spatial resolution. Under arctic conditions, the sensitivities are clearly reduced for SST and the SSS, as compared to warmer conditions. This will make the retrieval of these variables more challenging (leading to larger uncertainties) under cold environments. This is due to intrinsic physical changes in the dielectric properties of the ocean waters, from cold to warm temperatures. In addition, under arctic conditions, the frequency of the maximum sensitivity to SST and SSS significantly decreases. The maximum sensitivity to SST is between 2 and 4 GHz in the Arctic, and between 5 and 7 GHz in the Tropics. For

the SSS, using a flat ocean surface, Dinnat et al. (2018) and Le Vine and Dinnat (2020) observed a maximum SSS sensitivity at $\sim$400 MHz (resp. 800 MHz) at SST of $0°$ C (resp. $30°$ C). For the OWS, the sensitivity decreases with increasing atmospheric opacity, from the Arctic to the Tropics.

### 3.3 Sensitivities relative to the CIMR user requirement precisions for the Arctic

For the CIMR mission, the primary user requirements, as expressed in Donlon and CIMR Mission Advisory Group (2020)

in terms of standard total uncertainty, are $\leq$0.3 K for SST in polar regions ($55°$ N or S and above), and $\leq$0.2 K for global coverage, 5% for SIC (averaged over all seasons). For SSS, the requirement is $\leq$0.3 psu over monthly time-scales. Here, we calculate the change in measured $TB_{TOA}$ corresponding to the required parameter precision. In order to estimate the parameter with the required precision using a single channel retrieval, the instrument noise for that channel will have to be below that level. Note in addition that in these sensitivity calculations, the other parameters are fixed, i.e., there is no uncertainty related to

them. For parameters that are not driving the CIMR design, the following precisions are considered: 1 m/s for OWS, 1 kg/m$^2$ for TCWV, and 20 g/m$^2$ for TCLW.

  Figure 4 shows the results for each parameter in an arctic environment. It highlights the challenge to reach the geophysical precision expected to comply with the mission requirements. For the SST and SSS, the change in measured $TB_{TOA}$ is respectively of 0.08 K per 0.2 K at 6.9 GHz, and 0.06 K per 0.2 psu at 1.4 GHz, meaning that an instrument noise lower than

170 these values is required on the CIMR measurement at those frequencies to retrieve the parameters with the target precision in the arctic conditions, with single channel algorithms. For the OWS, the sensitivity is around 1 K per 1 m/s, and for the SIC, the sensitivity is strong with respect to the target precision, with 8 to 4 K per 5%. The TCWV and TCLW show respectively a sensitivity of 1.0 K per 1 kg/m$^2$ at 18.7 GHz, and 1.05 K per 20 g/m$^2$ at 36.5 GHz. Retrieval methods have been developed to benefit from observations in multiple channels with CIMR (Kilic et al., 2018; Jimenez et al., submitted). By using these

175 multi-channels algorithms, it is possible to improve the retrieval precision compared to a single channel algorithm. However, this analysis still provide an indication of the challenges to reach the required retrieval precisions.

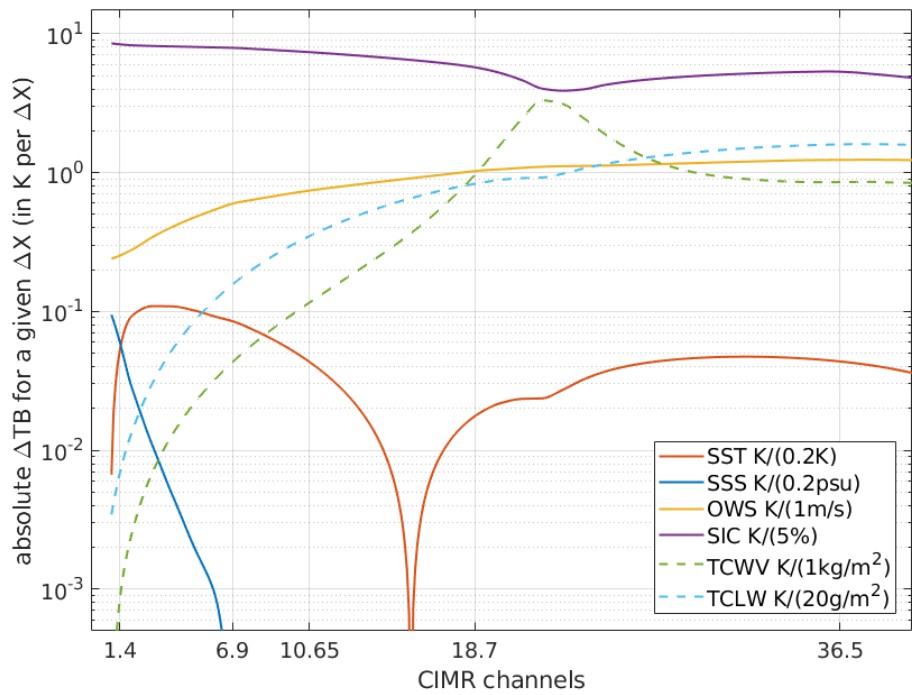

**Figure 4.** Changes in measured $TB_{TOA}$ corresponding to the required precision of the CIMR parameters. The sensitivities are not normalized and presented with a logarithmic scale for the y-axis. Calculations are performed as a function of frequency, at $55°$ incidence angle, in the Arctic (SST=$0°$ C, OWS=6 m/s, SSS=34 psu, TCWV=5 kg/m$^2$). The *units* are indicated in the legend for each parameter.

## 4 Conclusions

In this study, we computed the sensitivities to SST, SSS, SIC, OWS, TCVW, and TCLW as a function of frequency between 1 to 40 GHz, using the RTMs from Meissner and Wentz (2012) and Rosenkranz (2017) for the ocean and the atmosphere, and the RRDP for the sea ice. We improve the well-known figure from Wilheit (1979) with recent state-of-the-art radiative transfer models, taking into account the atmosphere, considering different geophysical conditions (arctic, mid-latitude, and tropical), and adding the sensitivity to the sea ice.

CIMR channels have been selected to provide the best compromise for ocean and sea ice products in terms of precision and spatial resolution. Our sensitivity analysis confirms the channel selection of CIMR: 1.4 GHz is the key frequency to estimate SSS, 6.9 GHz to estimate SST, and the channels between 6.9 and 36.5 GHz to estimate SIC. The frequencies from 18.7 to 36.5 GHz can provide information on OWS, TCWV, and TCLW. For the specific case of the arctic environment, the sensitivities to SST and SSS are smaller, and the maximum sensitivity is shifted toward the lower frequencies making the retrieval more difficult. However, the multiple channels of CIMR and their low instrument noises allow to satisfy the user requirements.

CIMR channels will ensure the continuity of the measurements from previous satellite microwave radiometers by considering the other constraints such as the full coverage of the poles, same orbit altitude as MetOp-SG, ITT frequency regulation, and possible RFI contaminations.

CIMR will provide simultaneous polarized measurements at 1.4, 6.9, 10.65, 18.7, and 36.5 GHz for the first time on a single satellite. The use of these multiple channels in coincidence to retrieve the ocean and sea ice surface parameters will be a major advantage to reach the target precision required by the user communities, especially in the polar regions. A first evaluation of the CIMR performances using a multi-channel analysis has been presented in Kilic et al. (2018). Algorithms are currently under developments to fully exploit the CIMR channels and reach the best performances for the estimation of the ocean and sea ice parameters (Kilic et al., 2018; Scarlat et al., 2020; Kilic et al., 2020; Prigent et al., 2020; Jimenez et al., submitted).

*Author contributions.* This study was conducted by L.K., and C.P.. C.J., and C.D. have contributed to the writting of the paper.

*Competing interests.* The authors declare no conflict of interest.

*Acknowledgements.* We are very grateful to Thomas Wilheit for his comments on his original figure. We thank Remote Sensing Systems for providing their ocean Radiative Transfer Model. We also acknowledge all the members of the CIMR Mission Advisory Group for their discussions and developments on the CIMR mission. This work is in part funded by the European Space Agency CIMR-APE study ESA Contract 40000125255.

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
