# Peer review of "Technical note: A sensitivity analysis from 1 to 40 GHz for observing the Arctic Ocean with the Copernicus Imaging Microwave Radiometer"

_Ocean Science, 2020_

## Referee Comment (RC1) · Anonymous Referee #1 · 17 Nov 2020

General comments

Firstly, I'd like to thank the authors for their work. I have marked for major revision as I think more is needed to address the benefits arising from having the CIMR channels on one platform. I did not feel the paper addressed this, despite saying it was very important.

The paper updates a popular figure that has been used for many years to illustrate the sensitivity of low frequency microwave observations to a range of geophysical products. The major change is that the new plot takes account of the atmosphere and adds SIC and additional plots show difference between tropics, middle latitudes and polar

latitudes and details for the arctic. The updated plot and new plots will be of interest but the paper does not give any particular new OS insights. It's a short paper to revisit the plot and look at sensitivity in the context of CIMR. It feels to me to be a first useful step in a wider study that will give a deeper understanding, but more progress is needed to justify publication.

Scientific significance: Fair

The paper does not reveal any substantial new understanding of ocean science.

Scientific quality: Good

The paper addresses the value of CIMR by examining the sensitivity of microwave frequencies to a range of geophysical parameters. Scientifically this is fine and the calculations are state of the art. However, it could perhaps have addressed more the linkage between frequencies. The paper says this is important, but then does not address this aspect.

Presentation quality: Good

The results are clear, but presenting only normalised sensitivities can give a misleading impression of the relative importance of sensitivities at a particular frequency, it only shows well the point of maximum sensitivity for each geophysical parameter. This is discussed further below.

Specific points requested by OS

1. Does the paper address relevant scientific questions within the scope of OS? CIMR is an important mission for OS, so the question is relevant.

2. Does the paper present novel concepts, ideas, tools, or data? The paper updates the work of Wilheit, but is not novel.

3. Are substantial conclusions reached? The variability with region is shown to be significant, but I think this is well known since papers such as Phalippou (1996) and of

Interactive
comment

course earlier work by Dr Prigent amongst others.

4. Are the scientific methods and assumptions valid and clearly outlined? Yes, though I would have liked to see more to address linkage between the frequencies, as this is a key aspect of CIMR, having 1 to 40 GHz on the same platform.

5. Are the results sufficient to support the interpretations and conclusions? Yes

6. Is the description of experiments and calculations sufficiently complete and precise to allow their reproduction by fellow scientists (traceability of results)? Yes

7. Do the authors give proper credit to related work and clearly indicate their own new/original contribution? Yes, given the limited ambition of the paper, though a more complete analysis of the MIMR, AMSR-E and AMSR2 literature is relevant, as CIMR only adds the L-band channel to the capability of these sensors.

8. Does the title clearly reflect the contents of the paper? It is not obvious to me why the authors choose to imply their paper is about the arctic. Fig. 2 is middle latitude, Fig. 3 compares arctic, middle latitude and tropics. Fig. 4 is arctic with a couple of paragraphs of discussion. Yes, SIC has been added compared to Wilheit. But overall there does not seem to be a particular emphasis on the arctic in the analysis. It's a global study.

9. Does the abstract provide a concise and complete summary? Yes

10. Is the overall presentation well structured and clear? Yes

11. Is the language fluent and precise? Yes

12. Are mathematical formulae, symbols, abbreviations, and units correctly defined and used? Yes

13. Should any parts of the paper (text, formulae, figures, tables) be clarified, reduced, combined, or eliminated? No
14. Are the number and quality of references appropriate? Yes – but more on MIMR etc literature would be appropriate, as noted above.

15. Is the amount and quality of supplementary material appropriate? No supplementary material needed

Specific comments

P2 L28 "No gap in coverage at the pole" What the authors mean is every orbit will see the pole, so there will be an observation every c.100 minutes. No gap may mislead some readers who are not familiar with polar orbiting satellites.

P2 L31 Not sure why you use the word harsh. It will observe all aspects of the arctic environment. This seems poetic language for a scientific paper.

P2 L33 Could you be more specific on the range of terrestrial products that CIMR will improve analysis of, that are additional to those you have just listed. Its not clear what you are talking about.

Equation 1: You have assumed a specular reflection. How realistic is this for some of the snow and ice surfaces you are concerned with? Suggest to clarify.

Equation 2: Why do you use finite difference rather than differentiating the code and how have you ensured that your dx is appropriate to get a robust estimate of the local gradient?

Equation 3: Whilst normalising like this maintains consistency with the Wilheit figure it may give a misleading impression of relative sensitivity to different parameters at a given frequency. It would be useful also to show the unnormalized figures.

Figure 4: I do not understand this figure. What is the cause of the sharp spectral feature in the SST sensitivity around 15 GHz? This makes no physical sense to me. Please explain.

P8 L145-155 I am struggling to see the point in this analysis. You have a multi-channel

instrument, you have already stated that the multi-channel aspect is important, its clear that single channel frequency retrievals are useless (even ignoring sensitivity to other parameters). I am not sure how this analysis gives new insights? It seems far less useful than a multi-channel information content study.

P9 L164-166 This does not seem to be a new finding, yet it reads like it is.

P9 L169-171 I agree the major aspect of CIMR is to use these frequencies together but this is really not explored in this short paper. The paper only repeats rather well known aspects of the sensitivity of individual frequencies. It is not difficult with the calculated gradients to explore the multi-channel aspects using linear information content theory. Its not new, but it would give more insight into the use of these channels together.

---

## Referee Comment (RC2) · Anonymous Referee #2 · 6 Dec 2020

This technical note deals with the sensitivity of the future CIMR microwave mission to various ocean and ice parameters. It is an update of the Wilheit figure that has been widely used with a focus on the incidence angle of CIMR (55°) and using a more recent modelling. I have no doubt that this information will be useful to the CIMR community, but I find the novelty of the paper quite modest with respect to other studies. I think with some changes (see below) the paper could represent a more important contribution to the community. It is nicely written and easy to read.

A first concern is about the atmospheric contribution : to which extent is the Rosenkranz (2017) model valid at L-Band ? The Rosenkranz citation corresponds to

a code and I did not find easily the corresponding references in the litterature but I am not sure at all it considers the contribution of the molecular Oxygen which is the dominant contribution at low frequency. Even though this contribution is much less than at higher frequency, given the low sensitivity of the brightness temperature to the salinity at L-Band, it cannot be ignored. This model is not used in salinity retrieval processors today.

Another concern is with Figure 2 : the title of the paper makes a focus on the Arctic Ocean but this figure is for the middle latitudes, I suggest you change the title of the paper or do this Figure for Arctic conditions. The conditions described in Table 2 are very restrictive and do not represent the true variation of the parameters. I suggest you consider more representative variations and report the corresponding sensitivity as a shadowing around median conditions on Figure 2. The normalisation of Figure 2 does not allow to get quantitative estimates. I suggest you add several Y axis with scales corresponding to the non normalized sensitivity for each parameter.

Detailed remarks :

Abstract Line 7-8 'state of the art': Levine and Dinnat (2020) recently published a similar study with a discussion of the sensitivity given by various state of the art ocean RTMs. The originality here is not to reproduce the figure of Wilheit with a recent model, but to update it at $55°$ which was not specifically studied by Levine and Dinnat. In addition, to my knowledge, the Rosenkranz atmospheric model is not considered as a state of the art model at L-Band.

Le Vine, D.M.; Dinnat, E.P. The Multifrequency Future for Remote Sensing of Sea Surface Salinity from Space. Remote Sens. 2020, 12, 1381.

Line 29-30 : what are the main parameters of interest on METOP-SG for CIMR ? Maybe add a reference for METOP-SG. What do the acronyms MWI/ICI and SCA mean ?

Line 40 : I guess it is meant : state of the art of the various components of Radiative

Transfer Model. In fact only one model is considered for each contribution whereas several are in use in the community and none of them have been absolutely ruled out given present uncertainties. I suggest to refer to the recent study of Levine and Dinnat who showed the sensitivities obtained with various widely used components of radiative transfer models.

Table 2 : how do realistic variations in the surface and atmospheric conditions modify the results ? I would suggest putting a shadow around the curves on Figure 2 to reflect the variations due to surface and atmospheric conditions as well as uncertainties coming from uncertainties on RTM.

Lines 68-75 : Molecular Oxygen is the main contributor at L-Band and is a significant contributor at low frequency (see Levine and Dinnat, appendix C), it should be considered. How does the Rosenkranz model compare with the MPM92 model more widely used at L-Band ?

Strictly speaking, equation 1 should be vertically integrated ; I guess neglecting the vertical integration might have some impact on the result, especially at high frequency, this should be discussed.

Line 91 and Figure 2: I guess you mean : maximum value of the sensitivity. I don't like much this normalisation because it does not allow a quantitative reading (I also have this problem when reading the Wilheit figure). You might envisage to add several Y axis with scales corresponding to the non normalized sensitivity for each parameter.

Line 111 : Partly redundant with the introduction

Legend of Figure 4 : unclear what does 'units' mean
* * *

---

## Referee Comment (RC3) · Anonymous Referee #3 · 7 Dec 2020

This technical note presents a sensitivity analysis of geophysical parameters relevant to the frequencies and viewing geometry anticipated for the arctic-focused CIMR mission. The focus of the work is a more quantitative reproduction of the classic Wilheit 1979 figure using more up-to-date information including RTM and inclusion of the atmospheric contribution. The note finishes with anticipated single channel TB sensitivity to the CIMR precision range for desired retrievables. The paper is well written and the results reasonably well presented. The information is highly useful for the CIMR team and future users of the data. The paper is lacking in scientific novelty for a publication. Suggestions are given below for some possible expansions that could add to the contribution of this work.

[Figure]

Specific Comments: Line 65: What are some of the uncertainties and sources of error associated with this?

Line 96: I understand the desire to present a single case here with explanation before diving in to the comparisons, but using the midlatitude case seems an odd choice here. I suggest reorganizing this section and either presenting 3 versions of Figure 2 (one for each of midlatitude/arctic/tropical) or (perhaps more in line with the original idea) leave the first section as "a general case" but average all areas together, making it truly general for this discussion, then next presenting Figure 3 teasing them all apart and discussing the latitudinal differences.

Lines 106-109: Is this the best place for this discussion? May be better placed in the introduction?

Section 3 - General: Please add a sense of the variability in these parameters, and how this variability differs for each of tropical/mid-lat/arctic. This is an important piece that is missing to make this more robust.

Section 3 - General: More discussion of differences from the Wilheit figure and how the changes are related to the updated technique and inclusion of the atmospheric contribution (more than the quick mention at line 101) would be a nice addition to this work.

---

## Author Comment (AC1) · 14 Jan 2021

First, we thank the reviewer for reading our paper and for his/her comments.

General comments Firstly, I'd like to thank the authors for their work. I have marked for major revision as I think more is needed to address the benefits arising from having the CIMR channels on one platform. I did not feel the paper addressed this, despite saying it was very important. The paper updates a popular figure that has been used for many years to illustrate the sensitivity of low frequency microwave observations to a range of geophysical products. The major change is that the new plot takes account of the atmosphere and adds SIC and additional plots show difference between tropics,

middle latitudes and polar latitudes and details for the arctic. The updated plot and new plots will be of interest butt he paper does not give any particular new OS insights. It's a short paper to revisit the plot and look at sensitivity in the context of CIMR. It feels to me to be a first useful step in a wider study that will give a deeper understanding, but more progress is needed to justify publication.

The CIMR mission has been described with more details in a previous publication (Kilic et al., JGR, 2018) and other papers describe some potential applications of CIMR (see for instance a list of publication at https://cimr.eu). The famous Wilheit figures has been used widely by the community, for CIMR and other missions. It was felt necessary to'officially' update this figure and to provide a reference for this update. This is the goal of this technical note. We are aware that this is not a full detailed study and this is the reason why we chose the technical note format.

Scientific significance: Fair The paper does not reveal any substantial new understanding of ocean science.

Scientific quality: Good

The paper addresses the value of CIMR by examining the sensitivity of microwave frequencies to a range of geophysical parameters. Scientifically this is fine and the calculations are state of the art. However, it could perhaps have addressed more the linkage between frequencies. The paper says this is important, but then does not address this aspect.

Presentation quality: Good The results are clear, but presenting only normalised sensitivities can give a misleading impression of the relative importance of sensitivities at a particular frequency, it only shows well the point of maximum sensitivity for each geophysical parameter. This is discussed further below.

Specific points requested by OS

1. Does the paper address relevant scientific questions within the scope of OS? CIMR

is an important mission for OS, so the question is relevant.

2. Does the paper present novel concepts, ideas, tools, or data? The paper updates the work of Wilheit, but is not novel.

3. Are substantial conclusions reached? The variability with region is shown to be significant, but I think this is well known since papers such as Phalippou (1996) and of course earlier work by Dr Prigent amongst others.

4. Are the scientific methods and assumptions valid and clearly outlined? Yes, though I would have liked to see more to address linkage between the frequencies, as this is a key aspect of CIMR, having 1 to 40 GHz on the same platform.

5. Are the results sufficient to support the interpretations and conclusions? Yes

6. Is the description of experiments and calculations sufficiently complete and precise to allow their reproduction by fellow scientists (traceability of results)? Yes

7. Do the authors give proper credit to related work and clearly indicate their own new/original contribution? Yes, given the limited ambition of the paper, though a more complete analysis of the MIMR, AMSR-E and AMSR2 literature is relevant, as CIMR only adds the L-band channel to the capability of these sensors.

8. Does the title clearly reflect the contents of the paper? It is not obvious to me why the authors choose to imply their paper is about the arctic. Fig. 2 is middle latitude,Fig. 3 compares arctic, middle latitude and tropics. Fig. 4 is arctic with a couple of paragraphs of discussion. Yes, SIC has been added compared to Wilheit. But overall there does not seem to be a particular emphasis on the arctic in the analysis. It's a global study.

The initial Wilheit figure was about open ocean at mid latitude. We added arctic simulations and sensitivity to sea ice.

9. Does the abstract provide a concise and complete summary? Yes

10. Is the overall presentation well structured and clear? Yes

11. Is the language fluent and precise? Yes

12. Are mathematical formulae, symbols, abbreviations, and units correctly defined and used? Yes

13. Should any parts of the paper (text, formulae, figures, tables) be clarified, reduced,combined, or eliminated? No

14. Are the number and quality of references appropriate? Yes – but more on MIMR etc literature would be appropriate, as noted above.

A reference to MIMR has been added.

15. Is the amount and quality of supplementary material appropriate? No supplementary material needed

Specific comments

P2 L28 "No gap in coverage at the pole" What the authors mean is every orbit will see the pole, so there will be an observation every c.100 minutes. No gap may mislead some readers who are not familiar with polar orbiting satellites.

The sentence has been clarified. It has been replaced by "CIMR . . . will fully cover the poles".

P2 L31 Not sure why you use the word harsh. It will observe all aspects of the arctic environment. This seems poetic language for a scientific paper.

"harsh" has been deleted

P2 L33 Could you be more specific on the range of terrestrial products that CIMR will improve analysis of, that are additional to those you have just listed. Its not clear what you are talking about.

Some precisions have been added "(e.g. soil moisture, vegetation dynamics, snow water equivalent )"

Equation 1: You have assumed a specular reflection. How realistic is this for some of the snow and ice surfaces you are concerned with? Suggest to clarify.

As demonstrated in Matzler (2005), the distinction between specular and lambertian scattering is not an issue for conically scanning instruments such as CIMR with incidence angles close to 55°. The problem arises for incidence angle close to nadir as can be the case for cross-track sounders such as AMSU or MHS.

Matzler, C. (2005). On the determination of surface emissivity from satellite observations. IEEE Geoscience and remote sensing letters, 2(2), 160-163.

Equation 2: Why do you use finite difference rather than differentiating the code and how have you ensured that your dx is appropriate to get a robust estimate of the local gradient?

In the microwave windows between 1 and 40 GHz, the variations of Tb as a function of the different parameters are quasi-linear, therefore we can use the finite difference to compute the sensitivity and the choice of dx is not very critical (contrarily to what happens in the calculation of the gradients in spectral lines).

Equation 3: Whilst normalising like this maintains consistency with the Wilheit figureit may give a misleading impression of relative sensitivity to different parameters at agiven frequency. It would be useful also to show the unnormalized figures.

Figure 4 is unnormalized with sensitivities that are plotted with logarithmic scale.

Figure 4: I do not understand this figure. What is the cause of the sharp spectralfeature in the SST sensitivity around 15 GHz? This makes no physical sense to me.Please explain.

This is because the results are presented with a logarithmic scale, because the sensitivities are very different depending on the parameters. Around 15 GHz the sensitivity to SST becomes zero, this is why we have this shape.

P8 L145-155 I am struggling to see the point in this analysis. You have a multi-channel instrument, you have already stated that the multi-channel aspect is important, its clear that single channel frequency retrievals are useless (even ignoring sensitivity to other parameters). I am not sure how this analysis gives new insights? It seems far less useful than a multi-channel information content study.

The multi-channel analysis has been done in Kilic et al., 2018. In this study, our goal is to present, in a convenient way for the users, the CIMR channels individually and their advantages in terms of sensitivity.

P9 L164-166 This does not seem to be a new finding, yet it reads like it is.

It is a confirmation and it summarizes to the community why these channels have been selected for CIMR.

P9 L169-171 I agree the major aspect of CIMR is to use these frequencies together but this is really not explored in this short paper. The paper only repeats rather well known aspects of the sensitivity of individual frequencies. It is not difficult with the calculated gradients to explore the multi-channel aspects using linear information content theory. Its not new, but it would give more insight into the use of these channels together.

As mentionned above, this multi-channel analysis has been presented in Kilic et al., 2018. Kilic, L., Prigent, C., Aires, F., Boutin, J., Heygster, G., Tonboe, R. T., ... & Donlon, C. (2018). Expected performances of the Copernicus Imaging Microwave Radiometer (CIMR) for an all‐weather and high spatial resolution estimation of ocean and sea ice parameters. Journal of Geophysical Research: Oceans, 123(10), 7564-7580.

Please also note the supplement to this comment:
https://os.copernicus.org/preprints/os-2020-92/os-2020-92-AC1-supplement.pdf

---

## Author Comment (AC2) · 14 Jan 2021

First, we thank the reviewer for reading our paper and for his/her comments.

This technical note deals with the sensitivity of the future CIMR microwave mission to various ocean and ice parameters. It is an update of the Wilheit figure that has been widely used with a focus on the incidence angle of CIMR (55°) and using a more recent modelling. I have no doubt that this information will be useful to the CIMR community, but I find the novelty of the paper quite modest with respect to other studies. I think with some changes (see below) the paper could represent a more important contribution to the community. It is nicely written and easy to read.

[Figure]

We are aware that this paper does not present fondamental novel results. It is a practical update of the Wilheit figure that has been widely used by the community. This is why we chose to submit this result as a technical note and not as a regular paper.

A first concern is about the atmospheric contribution : to which extent is the Rosenkranz (2017) model valid at L-Band ? The Rosenkranz citation corresponds to a code and I did not find easily the corresponding references in the litterature but I am not sure at all it considers the contribution of the molecular Oxygen which is the dominant contribution at low frequency. Even though this contribution is much less than at higher frequency, given the low sensitivity of the brightness temperature to the salinity at L-Band, it cannot be ignored. This model is not used in salinity retrieval processors today.

The Rosenkranz gas absorption coefficients have been widely used in the microwave community, even for the retrieval of water vapor and temperature in the assimilation of microwave satellite data in operational weather centers. It includes all the physics required for an accurate evaluation of the atmospheric absorption by water vapor and oxygen in the atmosphere. It is valid from 1 GHz up to 1000 GHz. The provided citation includes many more citations to different works from Rosenkranz and colleagues. The reviewer may refer to the MPM model from Liebe that is also widely used. Note that Liebe and Rosenkranz worked a lot together, with Rosenkranz still active, with the model we used including the latest updates.

Another concern is with Figure 2 : the title of the paper makes a focus on the Arctic Ocean but this figure is for the middle latitudes, I suggest you change the title of the paper or do this Figure for Arctic conditions. The conditions described in Table 2 are very restrictive and do not represent the true variation of the parameters. I suggest you consider more representative variations and report the corresponding sensitivity as a shadowing around median conditions on Figure 2. The normalisation of Figure 2 does not allow to get quantitative estimates. I suggest you add several Y axis with scales corresponding to the non normalized sensitivity for each parameter.
Figure 4 provides the information the reviewer suggests, without normalization.

Detailed remarks : Abstract Line 7-8 'state of the art': Levine and Dinnat (2020) recently published a similar study with a discussion of the sensitivity given by various state of the art ocean RTMs. The originality here is not to reproduce the figure of Wilheit with a recent model, but to update it at 55° which was not specifically studied by Levine and Dinnat. In addition, to my knowledge, the Rosenkranz atmospheric model is not considered as a state of the art model at L-Band. Le Vine, D.M.; Dinnat, E.P. The Multifrequency Future for Remote Sensing of Sea Surface Salinity from Space. Remote Sens. 2020, 12, 1381.

Thank you for the reference to the LeVine and Dinnat paper, we added it to our paper. The atmospheric model that is presented in LeVine and Dinnat is the MPM92 model from Liebe et al (1985, 1992) (reference 37 and 38 in there paper) with Rosenkranz improvements (reference 39 of the same paper). It is the same model that we use, but with a more recent reference.

Line 29-30 : what are the main parameters of interest on METOP-SG for CIMR ? Maybe add a reference for METOP-SG. What do the acronyms MWI/ICI and SCA mean ?

There are two instruments of interest on MetOp-SG, for synergy with CIMR. First the scatterometer ASCAT that can provide the ocean wind speed with accuracy, and second the two microwave imagers MicroWave Imager (MWI) and the Ice Cloud Imager (ICI) that extent the frequency range of CIMR up to 654 GHz, for atmospheric retrievals. Products from ASCAT (ocean wind speed) and from MWI/ICI (water vapor and liquid water content) could be used as first guess in the retrievals of CIMR.

The meanings of the acronyms have been added : "MetOp MicroWave Imager (MWI)/Ice Cloud Imager (ICI) and SCAtterometer (SCA) measurements"

Line 40 : I guess it is meant : state of the art of the various components of Radiative Transfer Model. In fact only one model is considered for each contribution whereas

several are in use in the community and none of them have been absolutely ruled out given present uncertainties. I suggest to refer to the recent study of Levine and Dinnat who showed the sensitivities obtained with various widely used components of radiative transfer models.

We agree that the paper from Levine and Dinnat presents an interesting comparison of the different radiative transfer models for the ocean. Our model selection is based on Kilic et al., 2019 that also compared different models (including the ones used in Levine and Dinnat), including comparisons with satellite observations. Kilic et al 2019 showed that the model that fits the observations better over the full frequency range and environment ranges is the model from RSS.

Table 2 : how do realistic variations in the surface and atmospheric conditions modify the results ? I would suggest putting a shadow around the curves on Figure 2 to reflect the variations due to surface and atmospheric conditions as well as uncertainties coming from uncertainties on RTM.

The variations due to the surface and atmospheric parameters are shown in Figure 3 for 3 typical cases. The uncertainties of the RTM is a problem that is partly treated in Kilic et al., 2019. We would like to keep Figure 2 easy to understand and close to Figure of Wilheit.

Lines 68-75 : Molecular Oxygen is the main contributor at L-Band and is a significant contributor at low frequency (see Levine and Dinnat, appendix C), it should be considered. How does the Rosenkranz model compare with the MPM92 model more widely used at L-Band ? Strictly speaking, equation 1 should be vertically integrated ; I guess neglecting the vertical integration might have some impact on the result, especially at high frequency, this should be discussed.

We are fully aware that O2 is the main contributor to the atmospheric absorption at L-band. See our response above. It is actually Rosenkranz who derived the oxygen parameter for the Liebe models. Liebe was mainly working on the water vapor attenuation. Liebe passed away several years ago and a new reference for the model is the one that is given in the paper.

Line 91 and Figure 2: I guess you mean : maximum value of the sensitivity. I don't like much this normalisation because it does not allow a quantitative reading (I also have this problem when reading the Wilheit figure). You might envisage to add several Y axis with scales corresponding to the non normalized sensitivity for each parameter.

We would like to keep Figure 2 close to the Wilheit one. But in Figure 4 the sensitivities are shown unnormalized with a logarithmic scale.

Line 111 : Partly redundant with the introduction

Yes the sentence "CIMR has a 55° incidence angle and a large swath (>1900 km) to provide full coverage of the poles (i.e., with no gap at the pole itself), for the first time with a conical scanner" has been deleted. We modified the following sentence by: "The choice of an incidence angle of 55° for CIMR has been constrained by the swath width (to fully cover the poles), and the spatial resolution."

Legend of Figure 4 : unclear what does 'units' mean

The units are the units of measurements for the sensitivities to the different parameters. It is explained in the text e.g., "The TCWV and TCLW show respectively a sensitivity of 1.0 K per 1 kg/m2 at 18.7 GHz, and 1.05 K per 20 g/m2 at 36.5 GHz".

Please also note the supplement to this comment:
https://os.copernicus.org/preprints/os-2020-92/os-2020-92-AC2-supplement.pdf

---

## Author Comment (AC3) · 14 Jan 2021

-First, we thank the reviewer for reading our paper and for his/her comments.

This technical note presents a sensitivity analysis of geophysical parameters relevant to the frequencies and viewing geometry anticipated for the arctic-focused CIMR mission. The focus of the work is a more quantitative reproduction of the classic Wilheit 1979 figure using more up-to-date information including RTM and inclusion of the atmospheric contribution. The note finishes with anticipated single channel TB sensitivity to the CIMR precision range for desired retrievables. The paper is well written and the results reasonably well presented. The information is highly useful for the CIMR team

and future users of the data. The paper is lacking in scientific novelty for a publication. Suggestions are given below for some possible expansions that could add to the contribution of this work.

-We are aware that this paper does not present fondamental novel results. It is a practical update of the Wilheit figure that has been widely used by the community. This is why we chose to submit this result as a technical note and not as a regular paper.

Specific Comments: Line 65: What are some of the uncertainties and sources of error associated with this?

-The sea ice emissivity varies upon different parameters the main one are the ice type, the ice thickness, and the snow depth. In the figure attached (Fig 1.) you can see the standard deviation of the ice brightness temperature (black rectangle) derived from the round robin data package that is cited in the paper.

Line 96: I understand the desire to present a single case here with explanation before diving in to the comparisons, but using the midlatitude case seems an odd choice here. I suggest reorganizing this section and either presenting 3 versions of Figure 2 (one for each of midlatitude/arctic/tropical) or (perhaps more in line with the original idea) leave the first section as "a general case" but average all areas together, making it truly general for this discussion, then next presenting Figure 3 teasing them all apart and discussing the latitudinal differences.

-We would like to keep Figure 2 close to the Wilheit initial figure for an easier comparison, before changing the geophysical condition.

Lines 106-109: Is this the best place for this discussion? May be better placed in the introduction?

-Yes, it has been moved to the introduction.

Section 3 - General: Please add a sense of the variability in these parameters, and how this variability differs for each of tropical/mid-lat/arctic. This is an important piece

that is missing to make this more robust.

-Yes we added sentences to explain the variation of the parameters: "The TCWV and the SST vary globally between roughly 5 to 60 kg/m2 and 273 to 305K, respectively, with mean values that strongly depend upon the latitude. The OWS and SSS vary globally between roughly 0 to 20 m/s and 32 to 38 psu with mode values around 7 m/s and 34 psu, respectively."

Section 3 - General: More discussion of differences from the Wilheit figure and how the changes are related to the updated technique and inclusion of the atmospheric contribution (more than the quick mention at line 101) would be a nice addition to this work.

-Yes, we added the following sentence: "Note that with this updated version of the Figure of Wilheit1979 taking into account the sensitivity to the atmosphere, we can see that the sensitivities to the other parameters, such as SST or OWS, are decreased at higher frequencies (>18 GHz), and especially near the water vapor absorption line at 22 GHz."

Please also note the supplement to this comment:
https://os.copernicus.org/preprints/os-2020-92/os-2020-92-AC3-supplement.pdf

———————————————

[Figure]

**Fig. 1.**